Determination of optical density (OD) of oligodeoxynucleotide from HPLC peak area

Chillar Komal
Yin Yipeng
Eriyagama Adikari M. Dhananjani N.
Fang Shiyue
Department of Chemistry, Michigan Technological University, Houghton, MI, United States
Author ContributionsKomal Chillar performed the experiments, analyzed the data, prepared figures and/or tables, authored or reviewed drafts of the article, and approved the final draft.

Yipeng Yin performed the experiments, authored or reviewed drafts of the article, and approved the final draft.

Adikari M. Dhananjani N Eriyagama performed the experiments, authored or reviewed drafts of the article, and approved the final draft.

Shiyue Fang conceived and designed the experiments, prepared figures and/or tables, authored or reviewed drafts of the article, and approved the final draft.

Corresponding author: Shiyue Fang, shifang@mtu.edu
Print publication date: 2022
Electronic publication date: 2022 Jul 7
Volume: 4
Electronic Location ID: e20
Author manuscript; available in PMC: 2022 Jul 17
License: Distributed under Creative Commons CC-BY 4.0
License URL: https://creativecommons.org/licenses/by/4.0/

Keywords: Analytical Chemistry (other), Novel Analytical Technologies, Sample Handling, Spectroscopic Analysis, UV-Visible Spectroscopy
Keywords: HPLC, OD, Oligonucleotide, Optical density, Quantification

==============================
Oligodeoxynucleotides (ODNs) are typically purified and analysed with HPLC equipped with a UV-Vis detector. Quantities of ODNs are usually determined using a UV-Vis spectrometer separately after HPLC, and are reported as optical density at 260 nm (OD260). Here, we describe a method for direct determination of OD260 of ODNs using the area of the peaks in HPLC profiles. It is expected that the method will save significant time for researchers in the area of nucleic acid research, and minimize the loss of oligonucleotide samples.

INTRODUCTION

Oligodeoxynucleotides (ODNs) synthesized on an automated synthesizer are usually purified using reversed-phase (RP) or ion-exchange HPLC (Beaucage & Herdewijn, 2021; Sinha & Jung, 2015). Their purity is also usually determined using HPLC. For both preparative and analytical HPLC, the elution profile is mostly generated using a UV-Vis detector with the wavelength set to 260 nm. Quantities of ODNs are typically documented using optical density at 260 nm, which is abbreviated as OD260. It is defined as the UV absorbance at 260 nm (A260) of the ODN to be quantified dissolved in 1 mL of water with a light path of 1 cm. The value of OD260 is usually determined separately after an ODN is purified by HPLC using an UV spectrometer or other UV based apparatus such as a NanoDrop. Because the peak area in an HPLC profile is a quantitative measure of the UV absorbance of the ODN eluted within the peak, we reasoned that a separate step for the determination of OD260 using a UV spectrometer as we usually do is unnecessary, and theOD260 value can be determined directly using the peak area in the HPLC profile. Here, we describe the establishment of a correlation curve between HPLC peak areas and OD260 values and demonstrate its use for the determination of OD260 using HPLC peak area without having to measure UV absorbance separately.

MATERIALS AND METHODS

Oligodeoxynucleotides (ODN) 1a (20-mer, 5′- TCA TTG CTG CTT AGA CCG CT-3′), 1b (21-mer, 5′-TTG CCA TGA TTG ACA ACC AAT-3′) and 1c (32-mer, 5′-TAG TTT TAT AAT TTC ATC AGC AGT GTT ACC GT-3′) were either obtained from a commercial source or synthesized on a MerMade-6 DNA/RNA synthesizer at 1 (1a) or 0.2 (1b and 1c) μmol scale under standard synthesis, cleavage and deprotection conditions (Fang, Fueangfung & Yuan, 2012). RP HPLC was carried out under typical conditions described elsewhere (Shahsavari et al., 2019). UV absorbance was obtained on a Horiba Scientific Duetta Fluorescence and Absorbance Spectrometer at 260 nm in a 1 mL quartz cuvette with a 1 cm light path. Water was used as the blank.

For establishing the correlation curve of OD260 vs HPLC peak area (Fig. 1, see supporting information for a protocol for establishing the curve), ODN 1a, which was synthesized at 1 μmol scale and purified with trityl-on RP HPLC, was dissolved in 1 mL water. Different volumes of the solution (see Table 1) were injected into HPLC. For each injection, the peak area of the ODN was recorded, and the fractions corresponding to the peak were collected, combined and concentrated to dryness in a centrifuge evaporator under vacuum. The ODN was dissolved in 1 mL water. UV absorbance at 260 nm, which is the OD260 of the ODN in the corresponding HPLC peak, was obtained on a UV-Vis spectrometer using the solution in a 1 mL cuvette with a 1 cm light path (Table 1). The correlation curve of the values of OD260 vs the HPLC peak areas for the injections were generated and presented in Fig. 1. The slopes of the line is 0.00033.

To demonstrate the use of the curve for the determination of OD260 from HPLC peak area, ODN 1b was synthesized at 0.2 μmol scale on CPG, cleaved, deprotected, and purified with trityl-on RP HPLC. One twentieth of the ODN was injected into HPLC. The area of the ODN peak was found to be 498.3, which corresponds to an OD260 of 0.164 according to the line obtained using the UV-Vis spectrometer in Fig. 1. Thus, the OD260 for the 0.2 μmol ODN synthesis was 3.29. To validate the result, the OD260 of the synthesis was also determined using standard method by measuring the absorbance at 260 nm on a UV-Vis spectrometer. The number obtained was 3.31. The use of the curve was further validated using ODN 1c following the same procedure for 1b. The OD260 values calculated from the line and obtained using a UV-Vis spectrometer were 7.42 and 7.10, respectively.

RESULTS AND DISCUSSION

Oligonucleotides including oligodeoxynucleotides (ODN) are usually quantified by measuring UV absorbance at 260 nm (OD260) in a separate step after HPLC purification or analysis. Once the value of OD260 of an oligonucleotide is obtained, its mass in micro grams or micro moles can be easily calculated based on its sequence. Such calculations are usually carried out using free online tools by simply imputing the sequence and the OD260 value. In this report, we describe a method to determine OD260 directly from the area of HPLC peak instead of obtaining the value in a separate step.

To use HPLC peak area to determine OD260, a correlation curve between OD260 and HPLC peak area needs to be established first (see supporting information for a protocol). ODN 1a is used to demonstrate the process. A solution of 1a from a 1 μmol synthesis was prepared (see Materials and Methods section). The concentration does not need to be known or accurate, but should be suitable for the generation of HPLC peaks with areas close to those in HPLC profiles a lab typically generates for ODN purification and analysis. For the case of 1a, the ODN from the 1 μmol synthesis was dissolved in 1 mL water. Various volumes as indicated in Table 1 were injected into HPLC. For each injection, the ODN under the correct peak was collected and the area of the same peak was recorded (Table 1). The ODN was evaporated to dryness, and dissolved in 1 mL of water. The value of OD260 was then obtained by measuring the absorbance using a UV spectrometer. Plotting the OD260 numbers against peak areas gave the required correlation curve (Fig. 1). As expected, the data fitted well with straight line with a slope of 0.00033.

Once the correlation curve is obtained, the OD260 for any ODN that has an HPLC profile can be easily calculated. The ODN 1b, which was synthesized at a 0.2 μmol scale, is used as an example. One twentieth of the crude ODN was injected into HPLC. The area of the ODN peak was found to be 498.3. Using the correlation line generated with the data from UV-Vis spectrometer in Fig. 1, the area corresponds to an OD260 of 0.164. Alternatively, the OD260 can be simply calculated using the slope of the line, which gave the same number (498.3 × 0.00033). With the number for a portion of the sample, the OD260 for the 0.2 μmol synthesis can be easily calculated, which is 3.29 (0.1644 × 20). To validate the result, the OD260 of the synthesis was measured in a standard way using a UV-Vis spectrometer, and the value was 3.31, which was close to the value calculated from the graph or the slope. The method was further validated using ODN 1c using the same procedure for 1b, and the numbers from the graph and from standard measurement were 7.10 and 7.42, respectively.

Although the method for the determination of OD260 is simple and convenient, it is recommended that the validity of the standard curve or slope is checked whenever the HPLC conditions such as column, eluents, gradient, flow rate and temperature are changed. In addition, if the flow cell of the UV detector is replaced or cleaned, a new curve should be generated. Gratifyingly, in most labs, HPLC is typically performed under consistent conditions and UV detector flow cell can last for many years, there is no need to validate the graph frequently. Another suggestion is that when the area of the peak of an ODN is out of the areas used to generate the correlation graph, caution is needed to use the graph to calculate its OD260 because the correlation may not be linear when the concentration of the ODN in the eluent is too high. We suggest only use HPLC peak area to determine OD260 when the concentration of the eluent is not too high and is within the linear curve range. Finally, it is important to point out that the usually long and thick waste eluent line connecting the UV detector flow cell to waste container needs to be removed when collecting fractions of eluent to generate the OD260-HPLC peak area correlation graph. If this were not followed, the fractions collected may not be the fractions corresponding to the intended HPLC peak.

CONCLUSION

In summary, a simple and convenient method for the determination of OD260 of oligonucleotides using HPLC peak area is described. Although only quantification of ODN is described here, it is conceivable that the method can be equally applicable to RNA quantification. Because synthetic oligonucleotides typically have to go through the HPLC procedure for the purposes of purification or analysis, the direct OD260 determination method can save time for researchers by bypassing the step of measuring UV absorbance using a spectrometer.

Supplementary Material

supporting information1

supporting information2

supporting information3

Funding

This work was supported by NSF (1954041) and NIH (GM109288). The funders had no role in study design, data collection and analysis, decision to publish, or preparation of the manuscript.

Grant Disclosures

The following grant information was disclosed by the authors: NSF: 1954041. NIH: GM109288.

Data Availability

The following information was supplied regarding data availability: The raw data are available in the Supplemental File.

Figure 1 The correlation curve of OD260 and HPLC peak areas. The slope of the line is 0.00033.

Table 1 HPLC peak areas and their corresponding OD260 valuesa.

Entry	Volume (μL)	HPLC peak area	OD260	
1	4.0	485.0	0.176	
2	6.0	781.3	0.269	
3	9.0	1,113.1	0.395	
4	12.0	1,497.6	0.496	
5	15.0	1,925.1	0.616	
6	20.0	2,408.3	0.778	
Note:

a The ODN 1a synthesized at 1 μmol scale was purified with trityl-on HPLC, and dissolved in 1 mL water. Different volumes were injected into HPLC. The peak areas were recorded, and the fractions corresponding to the peak areas were collected. The fractions were evaporated to dryness and dissolved in 1 mL water. The values of UV absorbance at 260 nm (OD260) were then obtained in a 1 mL cuvette with a 1 cm light path. The listed values were the average of three measurements.

Supporting information

Protocol for establishing a correlation curve between HPLC peak area and OD260.

Competing Interests

The authors declare that they have no competing interests.

Supplemental Information

Supplemental information for this article can be found online at http://dx.doi.org/10.7717/peerj-achem.20#supplemental-information.

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
