# Peer review of "Determination of optical density (OD) of oligodeoxynucleotide from HPLC peak area"

_PeerJ Analytical Chemistry, doi:10.7717/peerj-achem.20_

## Round 0.1 · original submission · Major Revisions

Thanks for the submission. As can be seen based on the comments from these three reviewers, the reported new HPLC approach for the direct measurement of DNA optical density can potentially be of interest in the analytical chemistry field. However, some further discussion/experiments on the accuracy of the method and necessary controls are still needed for the revised manuscript. We look forward to receiving your revision soon.

Reviewer 1 ·

Basic reporting

In this manuscript, they demonstrated a method for directly measuring the OD260 of ODNs using the HPLC peak area. They claimed that the method will save significant time and minimize the loss of oligonucleotide samples. However, they need to carefully check their accuracy and the affect of HPLC conditions on the standard curve. If they need to construct standard curve very often to get the accurate data, I did not see how this method will save significant time for researchers.
1. They should provide the original HPLC raw data. In addition, in Table 1 and Figure 1, they only performed once for each different sample. They should at least run triplicate samples to validate the repeatability.
2. For Figure 1, they should provide the standard curve equation. More importantly, they should expand their range of standard curves since they used the current standard curve to calculate the OD260 for ODN 1b. Obviously, the peak area of 1690.6 is out of the detection range.
3. To validate their method, they synthesize ODN 1b and 1c and compare the data obtained through the HPLC peak area and the data by measuring the absorbance at 260 nm on a UV-Vis spectrometer. While they only tested one sample for each ODN. They should at least test three samples for each by synthesizing at different scales. In addition, they should also calculate the accuracy to check whether they are in an acceptable range.
4. They claimed that by skipping the UV-Vis spectrometer measurement, they could be able to minimize the loss of oligonucleotide samples. While the volume of samples needed for Nanodrop is around 1-2 µL. Is this really significant to save this small amount of samples by risking the results may not be that accurate due to the poor quality of the standard curve that was caused by HPLC condition changes? They should explain more here.

Experimental design

no comment

Validity of the findings

no comment

Additional comments

no comment

Reviewer 2 ·

Basic reporting

Referencing is not appropriate and some original papers on DNA cleavage and purification needs to be cited. Currently, authors only cited their own their own previous works.

Experimental design

Experimental design is appropriate

Validity of the findings

Results seems to support their conclusions

Additional comments

In this paper, Chillar et al. reported a method to directly measure DNA optical density from their HPLC peak area. This is an interesting method to measure DNA concentration and can be potentially useful. I believe this paper should be eventually accepted to be published by Peer J Analytical Chemistry journal after authors answer following questions.

Authors used water as eluent and therefore blank in their measurement. However, for many post synthesis modifications, purification and quantification is done with gradient flow of organic and aqueous solutions. How this method can be adopted to these cases? Authors need to show method applicability and accuracy under such conditions.

Reviewer 3 ·

Basic reporting

Fang et al., developed a method to directly measure the concentration of oligonucleotides (ODNs) with the peak area obtained from the HPLC purification process. The authors established a linear correlation curve between the peak area and the absorbance of the same ODNs obtained from UV-Vis spectroscopy at 260 nm. With the correlation graph, the peak area of targeted ODNs could be directly converted to the absorbance at 260 nm. The authors tested three ODNs with different sequences and found the absorbances at 260 nm converted from the peak area aligned well with the ones from both UV-Vis spectrometer and Nanodrop. Overall, it is interesting to quantify the concentration of ODNs during the process of purification and save the step to separately measure it after drying the samples. However, there are several issues the authors need to address before publication:
1. The authors did not specify the mathematic approach they used to calculate the peak area and how accurate the calculation could be, which is quite important to the theory they claimed here.
2. The linear range of this method was not determined. Besides, the author claimed the concentration of the targeted ODNs is too high, the correlation between the peak area and the absorbance at 260 nm may not be linear and a simple solution to that was to collect fraction of the peak and add the datum to the graph. In this case, how accurate the graph can be used to determine the concentration of targeted ODNs with the peak area value in the nonlinear range?
3. There was only one data point for each injection volume and peak area included in Table 1 and Figure 1, and no error bars were included in Figure 1. More replicates need to be added to prove the reproducibility and consistence of the method.
4. The correlation coefficient (such as R2) of the two lines needs to be added in Figure 1.
5. The author mentioned the correlation graph between the peak area and the absorbance needed to be checked occasionally, especially when the column, eluents, or flow rate were changed. How long one correlation graph could be consistent to determine the absorbance of ODNs with the peak area? If all the parameters for the purification process were the same, yet the correlation graph could change, maybe due to day-to-day or sample-to-sample variations, would it be easier to measure the absorbance after drying the sample with UV-Vis spectroscopy?
6. It was indicated for ODN 1b and 1c, the absorbances at 260 nm obtained from the HPLC peak area and the UV spectrometer were quite different (1b is 12.16 and 11.20, while 1c is 16.92 and 15.40, respectively). The authors claimed it was possibly due to the loss between the transfer from HPLC eluent to UV measurements. What is the specific reason for the loss? Besides, does it mean the concentration of targeted ODNs obtained from the peak area would be different from the real final concentration since there was a loss in the process of solvent drying or centrifuge evaporation? How big the difference would be, and which result people should rely on?
7.Proper references need to be added to the introduction section.
8. The word “intuitively” from the last paragraph of the result section is not proper for a scientific article. Is there any specific reason or evidence that the fractions may not be the corresponding ones if without removal of the eluent collection tube?

Experimental design

no comment

Validity of the findings

no comment

---

## Round 0.2 · accepted · Accept

Thanks for the careful revision. Hope you and all the co-authors have found those comments from the reviewers helpful in improving the quality of the study. I am pleased to accept your manuscript in the current form. Congrats!

Reviewer 1 ·

Basic reporting

After the revision, the manuscript can be accepted now.

Experimental design

After the revision, the manuscript can be accepted now.

Validity of the findings

After the revision, the manuscript can be accepted now.

Reviewer 2 ·

Basic reporting

Clear and unambiguous, professional English used throughout

Experimental design

its properly designed and executed

Validity of the findings

Presented data supports their conclusions